# Occupational gender segregation and economic growth in U.S. local labor markets, 1980 through 2010

**William J. Scarborough** *

Department of Sociology, University of North Texas, Denton, TX, United States of America

* William.Scarborough@unt.edu

## Abstract

The exchange of diverse ideas has been shown to be a major driver of economic growth and innovation in local labor markets across the U.S. Yet, persistently high levels of occupational gender segregation pose a barrier to such exchange between women and men workers. Consistent with this, organizational sociologists have identified multiple economic benefits to gender diversity in workplaces. Yet, it is unclear whether these trends apply to local labor markets, which constitute the ecological geographic environment for firms. In this study, I use fixed effects regression models to examine the relationship between labor market levels of segregation and economic growth from 1980 through 2010. I find that gender segregation hinders the expansion of finance and technology sectors as two industries that rely on the exchange of information and innovation. Consequently, higher levels of gender segregation are also a bane to economic productivity, as measured through hourly wages. Results from this study suggest that gender equity, manifested in lower levels of occupational segregation, is a vital ingredient in the economic development of local U.S. labor markets.

## Introduction

Voluminous studies have explored the determinants of occupational gender segregation [1–6]. This body of work has identified how economic conditions [7,8], political regulatory environments [6,9], and cultural orientations [1,3,10] shape the extent to which women and men workers are employed in separate occupations. The common orientation in these works views occupational gender segregation as a consequence of economic and social forces. Viewed from a different perspective, however, occupational gender segregation may itself influence economic growth and change. Research on organizations has found that gender diverse work teams are more productive than homogenous ones [11–13] and that firms with more women leaders have better profits than firms where management is dominated by men [14–19]. These findings suggest that there are economic benefits to lower levels of occupational gender segregation. Yet, there is little research on the consequences of occupational gender segregation at the level of local labor markets. This is despite the fact that a major source of gender segregation occurs between-firms within geographical labor markets [20]. Additionally, economists

not have any special access privileges that others would not have in obtaining data.

**Funding:** The author received no specific funding for this work.

**Competing interests:** The author has declared that no competing interests exist.

and urban scholars have increasingly focused on local labor markets as crucial centers of economic growth [21–29]. These scholars argue that labor markets with conditions that facilitate the exchange of ideas between workers are best positioned for innovation and growth in the creative economy. Yet, this area of research has not considered gender or how occupational gender segregation poses a barrier to the exchange of ideas.

In this study, I explore whether levels of occupational gender segregation influence economic change in U.S. local labor markets between 1980 and 2010. First, I examine the relationship between gender segregation and the expansion of finance and technology sectors as areas of the economy that have had major impacts on the U.S. over the past 40 years. Second, I explore whether gender segregation is related to economic productivity, as captured by hourly wages. Finally, I test whether the relationship of finance and tech industry growth to hourly wages depends on levels of gender segregation. I investigate these three aspects using fixed effects regression equations that model concurrent change between occupational gender segregation and economic outcomes while controlling for unobserved stable characteristics of industries and labor markets. In testing for the causal mechanism by which gender segregation facilitates information exchange in high-skill finance and tech service sectors, I use retail trade as a counterfactual case where gender segregation would not operate through similar means to impact economic change. My results shed light on the economic costs of occupational gender segregation. Labor markets with higher levels of gender segregation have had lower levels of economic growth since 1980. Meanwhile, in areas of the U.S. where women and men more commonly work in the same occupations, finance and tech sectors have expanded alongside rising wages. These findings indicate that occupational gender integration boosts economic activity in high-skill service sectors through facilitating the exchange of ideas and information between women and men employees working in similar occupations.

## Labor market conditions for economic growth

Economists have long argued that the sharing of ideas and information facilitates economic innovation and growth [23,27,30–36]. Despite increased international trade and global integration, many scholars agree that innovation is fostered locally through networks of individuals possessing different types and forms of knowledge [23,30,34,37]. Critical to this process of endogenous development is the exchange of *diverse* types of knowledge and information. As Jovanovic and Rob ([33], pg. 570) state, "it is the combination of *different* ideas that gives rise to still *better* ideas" (emphasis theirs). Thus, a network of innovators with equal knowledge and experience will only "reinvent the wheel," while a network of innovators where each has unique knowledge and diverse experiences will produce something altogether new, different, and highly impactful.

Economists embracing this notion of endogenous growth have shed light on the social assets contained in labor markets that play a critical role in economic development [23,27,28,38–41]. Labor markets that foster the sharing of information through random meetings, chance encounters, and the promulgation of diverse perspectives have an advantage over labor markets with more rigid social structures where individuals are comparatively more isolated [23,27–28, 35,36]. To this extent, social capital is the driving force of contemporary economic growth. Innovation, rather than being born from the brilliance of single entrepreneurs, is an inherently social process facilitated by societal conditions within labor markets.

Urbanists have embraced the role of social settings in economic development to argue that economically thriving areas are those that have invested in fostering local cultures that promote creativity [25,27–29]. According to Florida [25,26], areas with hip nightlife, vibrant arts districts, and stimulating museums are best positioned for economic innovation and growth

because these features attract creative types of individuals and facilitate their interaction with one another–leading to an exchange of ideas and information. Additionally, Florida argues that one of the most central elements of local culture is tolerance towards sexuality and racial diversity. According to Florida [25], a culture that supports diversity is more attractive to creative individuals who are the drivers of tech industries. This finding is consistent with urban economists who have found positive associations between cultural diversity and economic productivity [22,27,28,42].

In short, economists point to the exchange of diverse ideas as a crucial driver of innovation, while urbanists argue that local cultures promoting diversity help attract creative individuals and foster inventive industries. Absent in these two bodies of literature is any consideration of how gender is implicated in the exchange of ideas that promote economic growth. It is a well-established cornerstone of feminist theory that women and men occupy different structural positions in society with distinct perspectives, worldviews, and standpoints [43,44]. Therefore, it would seem to reason that gender equity–through facilitating the interaction and equal collaboration of women and men–would fuel the exchange of ideas, unique information, and diverse perspectives. Yet, this asset has been given little attention in previous research, prompting the question *whither gender in the study of local labor market economic development*?

Existing answers to this question are limited to sociological studies examining the relationship between organizational diversity and firm performance. Within organizations, work teams with a more equal balance of women and men workers have been shown to have better sales, greater profits, and improved problem solving ability [11–13,16]. Women's representation in organizational leadership has also been found to have positive impacts within firms. Women's presence on corporate boards is associated with higher stock prices [15,18], while a greater proportion of women in management has been found to improve sales [14,17,19]. Beyond these direct benefits to economic performance, women's representation in firm leadership has also been linked to improved equity within companies through such measures as the increased representation of women and minorities, reduced gender pay gaps, and greater emphasis on programs around corporate responsibility [45–47].

At the organizational level, sociologists have established a strong relationship between gender diversity and economic performance. These studies illustrate the applicability of economic theories of endogenous growth at the organizational level through showing that gender diversity improves company output by facilitating the exchange of ideas and collaboration between equally situated women and men workers. Yet, theories of endogenous growth have focused primarily on geographical labor markets as sites of economic development. At this level of analysis, there is no corresponding literature on the relationship between gender diversity/gender segregation and economic performance. This is an important gap because local labor markets have emerged as a crucial site for economic development [21–29,41]. The exchange of ideas, information, and commerce between workers from various companies has been shown to be a major source of labor market growth [23,27,28,30–36]. Therefore, just as gender diversity boosts (and gender segregation hinders) firm performance, it is possible that similar trends may occur at the broader level of labor markets.

## Occupational gender segregation and labor market economies

Sociologists studying the U.S. have primarily examined gender diversity at the organizational level. Meanwhile, economists and urbanists have directed increased attention to local labor markets as crucial sites for growth and innovation. Therefore, in this study I explore whether gender diversity, measured inversely as the segregation of women and men into different occupations, predicts economic change.

My investigation is guided by three primary hypotheses. First, I examine whether occupational gender segregation is related to industry growth. Here, I focus on two sectors that have been found in previous literature to be associated with higher wages and economic productivity: finance and technology [48–52]. Because these sectors rely on the exchange of information and creative innovation more than other industries, I hypothesize that labor markets with higher levels of gender segregation will have less growth in finance and tech sectors than labor markets with lower levels of segregation.

*H1*: *Lower occupational gender segregation will be associated with expanding finance and tech sectors at the local labor market level.*

In addition to exploring industry expansion, I also examine whether gender segregation influences the *quality* of economic change as measured through hourly wages. I use hourly wages as a measure of economic wellbeing because it reflects the value and productivity of labor and because it is more closely tied to workers' improved economic standing. In testing the effects of gender segregation, I expect that higher occupational gender segregation will be related to diminished wage growth in finance and tech industries because it poses a barrier to the exchange of information and ideas that are key ingredients for economic productivity in these high-skill service sectors [30–36].

*H2*: *Lower occupational gender segregation will be associated with wage growth in the high-skill service sectors of finance and tech at the local labor market level.*

Finally, I explore whether gender segregation moderates the relationship of high-skill service sector expansion to wage growth. Because finance and tech sectors have been a major source of economic growth over the past 40 years [48–52], I examine whether their relationship to hourly wages depends on levels of occupational gender segregation. I anticipate that finance and tech sectors will be associated with improvements in hourly wages when accompanied by lower levels of occupational gender segregation that facilitate the exchange of information and improve productivity.

*H3*: *Finance and tech sector expansion will be associated with wage growth when accompanied by lower levels of occupational gender segregation at the local labor market level.*

## Research design

### Unit of analysis

This study focuses on change in local labor markets over time. Yet, multiple definitions have been used in previous research to define geographic labor markets [21,53,54]. For the purposes of this study, labor sheds defined by the Bureau of Economic Analysis provide an ideal measure of labor markets by prioritizing the containment of residents. More than other units used to define labor markets, labor sheds maximize the degree to which residents' commuting patterns are contained within geographic boundaries. In 2010, 93 percent of residents lived and worked within the boundaries of a labor shed, higher than the definitions for "labor market areas" (86 percent) or metropolitan statistical areas (80 percent) [53]. By virtue of prioritizing commuting zones, labor sheds often incorporate core based statistical areas and metro areas with high levels of integration. Because the hypothesized benefits of occupational gender segregation occur through the exchange of workers across occupations and firms within a shared geographic space, labor sheds constitute an ideal unit of analysis for this study by virtue of delineating the boundaries within which populations of employees work and reside. I use the most recent definitions provided by the Bureau of Economic Analysis in 2004 (see [53]) that demarcate 179 labor sheds

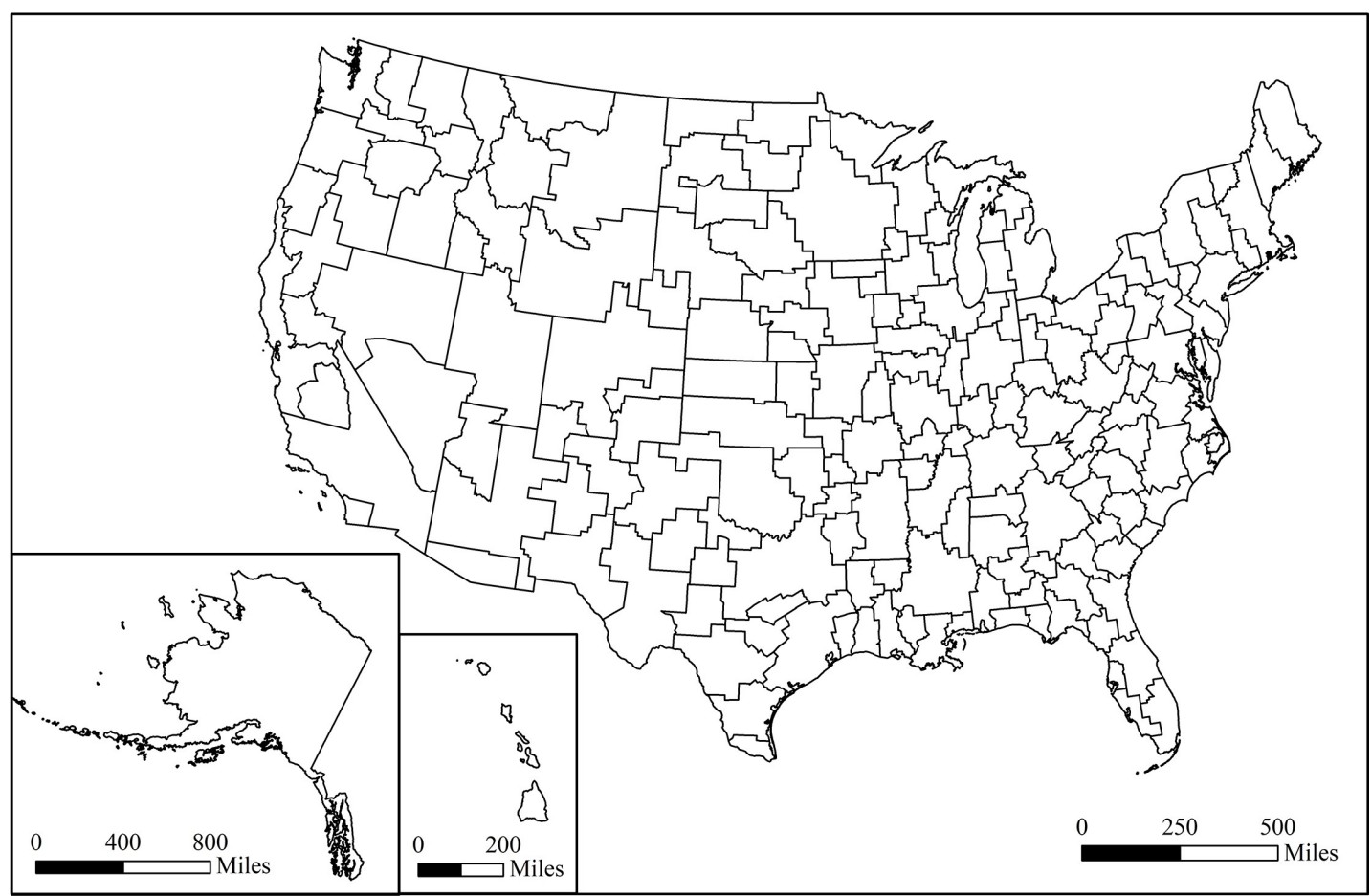

**Fig 1. U.S. local labor markets, defined as BEA labor sheds.**

across the U.S. Fig 1 maps the 179 labor sheds used in this analysis. These labor shed boundaries were used for all years included in the study (1980, 1990, 2000, and 2010). In total, the sample includes 716 labor shed-years. Henceforth, I refer to labor sheds as local labor markets.

## Data

The primary source of data for this study is the Integrated Public Use Microdata Series (IPUMS) [55]. For years 1980, 1990, and 2000, data come from the U.S. Census microdata series. For year 2010, I use the 2010 American Community Survey (ACS) 5-year microdata sample because the ACS replaced the U.S. Census's long form in the 2010 Census. I matched respondents to labor markets by geospatially matching PUMAs (Public Use Microdata Areas) (for years 1990, 2000, and 2010) and county groups (for year 1980) to labor markets. In instances where PUMAs/ county groups straddled the borders of labor markets, I adopted the approach used in previous research [21,54,56,57] and weighted respondents based on the likelihood of residing in each labor market. Further details on this method are included in S1 Appendix.

## Analytic approach

The hypotheses outlined above propose that decreasing occupational gender segregation will be associated with finance and tech sector growth as well as improved productivity measured

through wages. To test this relationship, I use fixed effects regression equations modelling the relationship between concurrent change in gender segregation and economic outcomes from 1980 through 2010. One benefit of fixed effects models is that they control for unobserved stable characteristics of labor markets and industries. One limitation, however, is that these equations are unable to account for reverse causation or control for unobserved time variant characteristics. While instrumental variable regression models are often used to address such problems of endogeneity, I was unable to find an appropriate instrumental variable to use in the context of this study. Instead, my focus on high-skill finance and tech industries provides an opportunity to use a counterfactual case of a low-skilled service industry where the effects of gender segregation on economic growth via information exchange would be absent, since low-skilled industries rely less on innovation and creative exchange. I use retail trade as a counterfactual. Not only are finance and tech workers twice as likely to have a college degree than those in retail trade, but the type of work in the retail trade sector is far less focused on creativity and innovation than that occurring in finance and tech [58]. In a sector where innovation is emphasized less, the economic benefits of occupational gender integration should be fewer. Examining retail trade as a counterfactual to finance and tech sectors provides a test for bias related to omitted variables and endogeneity to the extent that these confounding attributes would be present for retail trade to a similar degree as finance and tech. In other words, if occupational gender integration facilitates the exchange of ideas and innovation to further economic growth, we should see these patterns in industries where innovation is emphasized and rewarded (finance and tech) and not in industries where innovation plays a smaller role (retail trade).

In examining the first hypothesis related to industry growth, I use Eq 1 that predicts sector size with a set of industry and labor market characteristics varying by year.

$$x_{ilt} = \beta_0 + \beta_1 d_{ilt} + \beta_2 R_{ilt} + \beta_3 M_{lt} + \alpha_i + c_l + z_t + e_{ilt} \tag{1}$$

Industry size, $x_{ilt}$, is measured as a share of total labor market employment and varies by industry, labor market, and year. Economic sectors were identified using the harmonized longitudinal codes provided by IPUMS which offer consistent industry classifications from 1980 through 2010. Industry codes were standardized to those present in 1990 in this harmonized classification system. The finance sector includes industries related to finance, insurance, and real estate and is referred to using the acronym FIRE. The retail trade industry includes retailers and stores. To identify technological industries, I used Hecker's [59,60] classification system which identified tech industries as those with at least twice the proportion of technology-oriented occupations as the average for all industries. I include only technological service industries since this area pertains specifically to research, development, and information fields as opposed to tech manufacturing.

The focal predictor in Eq 1, $d_{ilt}$, is the index of dissimilarity measuring occupational gender segregation by industry, labor market, and year. $d$ is commonly used to measure occupational gender segregation [61,62] and captures the degree to which women and men workers are unevenly distributed across separate occupations with respect to their overall representation in the labor force:

$$d_{ilt} = \frac{1}{2} \sum_{i=1}^{n} \left| \frac{w_{oilt}}{W_{ilt}} - \frac{m_{oilt}}{M_{ilt}} \right| \tag{2}$$

Where segregation is measured as the summed absolute difference in women's ($w_{oilt}$) and men's ($m_{oilt}$) population across occupations relative to their overall population in the industry ($W_{ilt}$ and $M_{ilt}$, respectively). The resulting figure can be interpreted as the percentage of

women and men who would need to change occupations in order for them to be evenly distributed. A dissimilarity score of 1 indicates perfect segregation, while a score of 0 indicates perfect occupational gender integration. In general, scores above .6 are considered high levels of segregation, while those around .4 to .5 are considered moderate [63,64]. I used harmonized occupational codes from 1990 provided by IPUMS to ensure comparability of occupations across years. I calculated the dissimilarity index for FIRE, tech, and retail trade across each labor market-year in the sample.

Eq 1 also includes a number of covariates controlling for the changing characteristics of industries and labor markets. At the industry-level, $R_{ilt}$ includes the occupational racial segregation of white and black workers (measured with the index of dissimilarity), the percent of workers with a college degree, the share of employees that are women, and racial composition measured as the percent of workers who are white. Varying by labor market-years, $M_{lt}$ includes population (logged), the proportion of the population aged 25 through 40, racial composition (percent white), the unemployment rate, the share of the population with a college degree, the rate of women's labor force participation, the ratio of employment in service to goods producing industries as an indicator of deindustrialization, and the presence of international trade measured as the share of employment in export-intensive industries (identified by Riker [65]). Eq 1 also includes fixed effects for industry ($\alpha_i$), labor market ($c_l$), and year ($z_t$), accounting for stable characteristics of industries and labor markets as well as average differences between years.

In testing Hypothesis 1, I first apply Eq 1 to the full sample of industry-labor market-years to identify the general effects of gender segregation on industry growth across FIRE, tech, and retail trade. Then, to test whether occupational gender segregation has different effects by industry, I apply Eq 1 to three additional models examining each industry separately. I use this approach, as opposed to interaction terms, because certain covariates (e.g. percent college educated) have varying effects by industry. Across all models, standard errors are clustered by labor markets.

To examine hypotheses 2 and 3 related to economic productivity and wellbeing, I use the same basic model as Eq 1, but integrate sector size as a covariate and use median hourly wage as a dependent variable. I use median hourly wage of industry-labor market-years because my interest is on the general relationship between occupational gender segregation and industry productivity, rather than on the distribution of wages within a particular industry. I conducted additional robustness tests using wages at the 25[th] and 75[th] percentile to determine whether trends persist across the wage distribution. Findings were consistent with those presented here focusing on median wages. Results for these additional models are included in S2 Appendix. Hypothesis 2 is tested through examining the direct effects of occupational gender segregation on hourly wages in FIRE, tech, and retail trade. To examine Hypothesis 3, I add an interaction term testing whether the effects of industry growth on hourly wage differ by levels of occupational gender segregation.

$$y_{ilt} = \beta_0 + \beta_1 d_{ilt} + \beta_2 R_{ilt} + \beta_3 M_{lt} + \beta_4 x_{ilt} + \beta_5 (d_{ilt} \times x_{ilt}) + \alpha_i + c_l + z_t + e_{ilt} \qquad (3)$$

I use four sets of models to test whether occupational gender segregation impacts economic productivity and wellbeing. The first set of models explores the pooled sample of industry-labor market-years, while three subsequent sets examine FIRE, tech, and retail trade independently. As noted above, retail trade serves as a counterfactual to FIRE and tech. Because the exchange of information facilitated by occupational gender integration should benefit the high-skill sectors of FIRE and tech more than the low-skill sector of retail trade, I expect to

observe significant coefficients for occupational segregation when examining FIRE and tech sectors, and non-significant coefficients when testing relationships in retail trade.

In the following section, I first review basic descriptives highlighting gender segregation and economic change from 1980 to 2010. Then, I examine whether changing levels of occupational gender segregation predict industry expansion from 1980 through 2010. In the last section of the results, I test whether occupational gender segregation is related to economic productivity, as measured by hourly wages in FIRE, tech, and retail trade industries.

## Results

### Descriptives

Table 1 reports industry size, hourly wages, and occupational gender segregation by year across FIRE, tech, and retail trade. In general, all three industries have experienced growth across labor markets during the period of study. The tech sector has grown the largest as a proportion of its size in 1980. On average, tech services accounted for 4.3% of labor market employment in 1980 and 5.4% in 2010 –an increase of about 25%. The proportion of workers in FIRE has risen, but mostly between 1980 and 1990. Retail trade accounts for a much larger share of local employment than FIRE and tech, with an average of more than 15% of employees working in this industry. While there are substantive differences between FIRE, tech, and retail trade industries, the common trend of expansion across all three during the period of study suggests that they are collectively influenced by shared economic shifts occurring during the period of deindustrialization occurring after 1980.

Table 1 also reports indicators of economic productivity and wellbeing. Wages (in 2010 USD) in retail trade are between $2 and $7 an hour lower than FIRE, and between $9 and $11 lower than tech services across each decade. This is consistent with the fact that retail trade is a lower-skilled industry than FIRE and tech. Wages in retail trade were highest in 1980 ($12.16) and have been at lower levels ever since. Meanwhile, FIRE wages have increased with each decade. From 1980 to 2010, wages in FIRE increased by an average of $2.29 across labor markets. Wages in tech services were the highest of the sectors observed here, exceeding $20 an hour in each decade, but have experienced no growth, on average, since 1980.

Beyond economic shifts, there were also changes in the degree of occupational gender segregation. On average, gender segregation decreased across U.S. labor markets for each of the

**Table 1. Descriptive statistics of labor markets, 1980 to 2010.**

|  | 1980 | | | 1990 | | 2000 | | 2010 | |
|---|---|---|---|---|---|---|---|---|---|
|  | **Mean** | **SD** | | **Mean** | **SD** | **Mean** | **SD** | **Mean** | **SD** |
| *Sector Growth* | | | | | | | | | |
| % of Labor Force in FIRE | 5.12% | 0.013 | | 5.54% | 0.014 | 5.46% | 0.015 | 5.70% | 0.015 |
| % in Tech | 4.31% | 0.014 | | 4.48% | 0.013 | 5.33% | 0.017 | 5.39% | 0.016 |
| % in Retail Trade | 15.52% | 0.015 | | 16.71% | 0.015 | 16.63% | 0.014 | 17.19% | 0.014 |
| *Hourly Wage (2010 USD), by Sector* | | | | | | | | | |
| FIRE | $14.96 | 1.40 | | $15.15 | 2.01 | $16.67 | 2.59 | $17.25 | 2.90 |
| Tech | $21.30 | 2.48 | | $20.12 | 2.83 | $20.74 | 3.33 | $21.17 | 4.00 |
| Retail Trade | $12.16 | 1.15 | | $10.71 | 1.38 | $11.50 | 1.26 | $10.44 | 1.03 |
| *Occupational Gender Segregation by Sector* | | | | | | | | | |
| FIRE | 0.628 | 0.069 | | 0.544 | 0.082 | 0.486 | 0.080 | 0.443 | 0.079 |
| Tech | 0.721 | 0.083 | | 0.648 | 0.100 | 0.586 | 0.101 | 0.563 | 0.097 |
| Retail Trade | 0.504 | 0.038 | | 0.422 | 0.038 | 0.386 | 0.038 | 0.374 | 0.044 |

three industries analyzed here (see Table 1). In each decade, tech services had the highest, and retail trade the lowest, levels of occupational gender segregation. Geographic variation in occupational gender segregation across FIRE, tech, and retail trade industries in 2010 is illustrated in Fig 2. Across industries, middle-America has consistently high segregation, while the West Coast has consistently low segregation. Labor markets such as Atlanta, GA and Milwaukee, WI have higher relative segregation in FIRE industries than tech, while the opposite is true in labor markets like Miami, FL and Eugene, OR.

## Occupational gender segregation and industry growth

The descriptives indicate that there has been meaningful economic change as well as shifts in occupational gender segregation between 1980 and 2010. Although some aspects have changed more than others, and there is significant variation both between labor markets and within labor markets over time in the extent of change. To further examine the relationship between occupational gender segregation and economic change, I now present the results of fixed effects regression models predicting industry growth in Table 2.

The first column in Table 2 includes all industries in a pooled model to identify the general effect of occupational segregation on industry expansion, while the following models examine FIRE, tech services, and retail trade separately. The pooled model provides evidence that, in general, occupational gender segregation hinders industry expansion. Industries with higher levels of gender segregation have experienced less growth between 1980 and 2010 ($p < .001$). Conversely, lower levels of gender segregation predict industry expansion. Several covariates also influenced industry expansion in the pooled model, yet many of these predictors had differing effects by industry that were uncovered in subsequent models.

The second and third columns in Table 2 test Hypothesis 1 regarding the relationship between occupational gender segregation and the expansion of FIRE and tech sectors. The results indicate that occupational gender segregation plays a large role in these high-skill service industries. In both FIRE and tech, occupational gender segregation is negatively associated with growth ($p < .001$). Conversely stated, occupational gender integration–the tendency for women and men to work in similar occupations–predicts FIRE and tech expansion. The coefficients indicate that a change in industry levels of occupational gender segregation from highly segregated ($d = .6$) to moderately segregated ($d = .5$) is associated with growth of over .25 percentage points in the size of FIRE industries and about .39 percentage points in local tech sectors. Among covariates, the share of industry employees with a college degree is associated with growth across both sectors ($p < .05$). FIRE industries have grown the most in areas where women are well represented in this sector ($p < .001$) and where women's labor force participation is higher ($p < .05$). This is consistent with previous research showing that women have made major inroads in employment in finance [66]. FIRE growth has also occurred in labor markets with a younger working-aged ($p < .01$), college educated ($p < .01$), and majority white ($p < .01$) population, and where unemployment is higher ($p < .05$). Tech expansion, meanwhile, has occurred in labor markets with more global trade ($p < .01$) and a higher ratio of employment in service relative to goods producing industries ($p < .001$).

The fourth column in Table 2 examines predictors of retail trade expansion. As noted earlier, retail trade serves as a counterfactual because this low-skill service industry should not benefit from the exchange of information between women and men workers to a similar degree as FIRE and tech. If gender segregation affects FIRE and tech sectors through the hypothesized mechanisms, it should be unrelated to economic change in retail trade. Indeed, the results in Table 2 show that occupational gender segregation is unrelated to industry growth in retail trade, adding validity to the earlier findings for FIRE and tech. Among

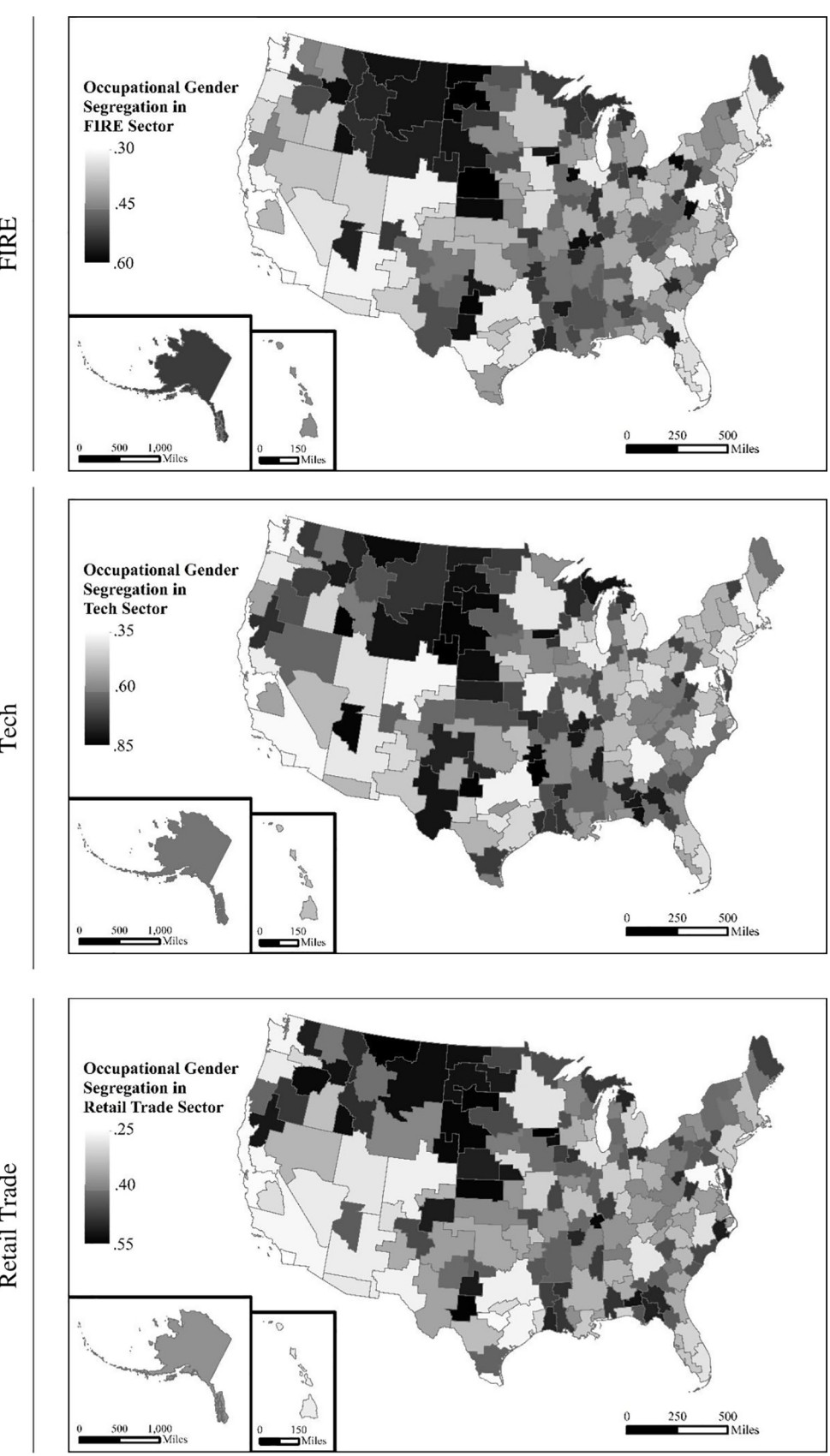

Fig 2. Occupational gender segregation across local labor markets in 2010.

covariates, areas with more women employed in retail trade experienced greater expansion of this sector (p < .01), as did labor markets with a larger share of white residents (p < .01), higher levels of unemployment (p < .001), and greater employment in service relative to goods

**Table 2. Fixed effects models predicting industry expansion (share of employment), 1980–2010.**

| | Pooled | FIRE | Tech | Retail Trade |
|---|---|---|---|---|
| Industry Characteristics | | | | |
| Occupational Gender Segregation | -6.666*** | -2.525*** | -3.887*** | 0.600 |
| | (0.776) | (0.576) | (0.785) | (1.372) |
| Occupational Racial Segregation | 0.155 | -0.004 | -0.137 | 0.033 |
| | (0.153) | (0.104) | (0.0976) | (0.253) |
| % College Educated | 8.243*** | 2.722* | 4.153* | 4.441 |
| | (1.235) | (1.093) | (1.928) | (6.142) |
| % Women | -1.064 | 2.844** | -1.454 | 5.928** |
| | (1.026) | (0.925) | (0.850) | (2.127) |
| % White | -3.983** | -2.117 | -3.075 | 0.430 |
| | (1.343) | (1.320) | (1.705) | (1.676) |
| Labor Market Characteristics | | | | |
| Population (logged) | -0.046 | -0.011 | -0.609 | 1.026 |
| | (0.282) | (0.385) | (0.446) | (0.586) |
| % Foreign Born | -4.685 | -2.365 | -2.586 | 0.561 |
| | (2.450) | (3.413) | (6.001) | (3.592) |
| % Aged 25–40 | -0.613 | 6.561** | 0.491 | -1.303 |
| | (2.496) | (2.360) | (4.663) | (3.750) |
| % White | 4.766*** | 4.740** | 0.643 | 6.336** |
| | (1.367) | (1.785) | (3.257) | (2.226) |
| Unemployment Rate | 4.253** | 5.498* | -4.430 | 11.692*** |
| | (1.420) | (2.167) | (2.278) | (2.998) |
| % College Educated | -5.766* | 9.497*** | 9.281 | -12.595* |
| | (2.562) | (2.356) | (4.987) | (4.990) |
| % Employed in Export-Intensive Industries | -2.987* | -2.413 | 5.260** | -16.543*** |
| | (1.163) | (1.439) | (1.809) | (1.916) |
| Women's Labor Force Participation | -0.023 | 3.797* | -2.995 | -1.765 |
| | (1.301) | (1.466) | (1.874) | (2.270) |
| Service to Goods Producing Employment Ratio | 0.246** | -0.064 | 0.474*** | 0.340** |
| | (0.0814) | (0.010) | (0.124) | (0.120) |
| Constant | 20.591*** | -2.031 | 15.829* | -2.267 |
| | (4.478) | (5.774) | (6.124) | (8.590) |
| Fixed Effects | Year Labor Market Industry | Year Labor Market | Year Labor Market | Year Labor Market |
| N | 2148 | 716 | 716 | 716 |
| R-sq | 0.962 | 0.433 | 0.580 | 0.662 |

*NOTE*: Dependent variable (share of employment) multiplied by 100. Standard errors clustered by labor market, reported in parentheses

*p<0.05

**p<0.01

***p<0.001

producing industries (p < .01). Labor markets with a higher proportion of college graduates (p < .05) and a larger share of workers in export-intensive industries (p < .001) experienced less growth in retail trade. This is consistent with the fact that these characteristics are associated with growth in high-skill industries.

In summary, the results from Table 2 support Hypothesis 1, that occupational gender segregation hinders growth in FIRE and tech sectors. This was supported in two ways. First, fixed effects models showed that gender segregation negatively predicts industry expansion in these two sectors. While this relationship could be due to unobserved changing characteristics of industries or labor markets, the counterfactual of retail trade–an industry that would be similarly affected by confounders but would not be affected in the same way by gender segregation as FIRE and tech industries, provided support that occupational gender integration affects high-skill service industries (FIRE and tech) through facilitating information exchange, while having no effect on low-skill service industries (retail trade) where such exchange is less beneficial to growth. It is notable that the coefficient for occupational gender segregation in the pooled model is greater in size than the coefficients observed in the models focusing on FIRE and tech sectors. This is because the pooled model does not account for the varying effects of controls across industries, while the subsequent equations with independent models by industry allow for this level of detail. For example, the share of local employment in export-intensive industries has no association with FIRE growth, a positive relationship with tech growth, and a negative relationship with growth in retail trade. Controlling for these types of varying effects mediated the coefficients for occupational segregation in the industry-specific models from what would be expected given its size in the pooled model. Nonetheless, the coefficients remain significant and substantial.

While these results indicate that gender segregation plays a large role in shaping the type of industry growth taking place in U.S. local labor markets, they provide little information on the quality of those economic developments and whether they improve productivity. The next section examines this aspect of economic change.

## Occupational gender segregation and economic productivity

Table 3 presents the results of models predicting change in hourly wages across FIRE, tech, and retail trade sectors. Each pair of models first tests the direct effects of occupational gender segregation on wage growth before exploring whether occupational gender segregation moderates the relationship of industry growth to wages.

Models 1 and 2 in Table 3 report the results from a pooled equation including all industries to identify the general effects of occupational gender segregation on wages. These models include a fixed effect term for industry and assume invariant effects of predictors across FIRE, tech, and retail trade. The first model shows that, in general, occupational gender segregation hinders wage growth (p < .001), while sector expansion is associated with an increase in local wages (p < .001). Among covariates, women's labor force participation rate (p < .001) and women's industry representation (p < .001) are both associated with lower wages, while the share of labor market (p < .001) and industry (p < .001) workers with a college degree, as well as the percent of workers in an industry who are white (p < .01), predict wage growth. The second model tests whether the relationship of industry expansion to wages depends on occupational gender segregation. Here, the findings for the pooled model indicate that sector expansion is more strongly associated with wage growth in highly segregated labor markets than those with lower levels of occupational gender segregation (p < .001). While these findings are counter to Hypothesis 3, they should be interpreted with caution because the pooled model does not control for the varying effects of control variables across industries or

**Table 3. Fixed effects models predicting labor market economic productivity as hourly wages, 1980–2010.**

| | Pooled Hourly Wage | | FIRE Hourly Wage | | Tech Hourly Wage | | Retail Trade Hourly Wage | |
|---|---|---|---|---|---|---|---|---|
| | Model 1 | Model 2 | Model 3 | Model 4 | Model 5 | Model 6 | Model 7 | Model 8 |
| Sector Characteristics | | | | | | | | |
| Occupational Gender Segregation | -5.169*** | -7.321*** | -2.441** | 5.185** | 0.360 | 7.440** | 1.439 | -5.072 |
| | (0.821) | (0.909) | (0.787) | (1.765) | (1.538) | (2.373) | (0.915) | (7.705) |
| % Employed in Sector | 0.242*** | 0.045 | 0.165* | 0.900*** | 0.372** | 1.320*** | -0.027 | -0.197 |
| | (0.035) | (0.051) | (0.078) | (0.155) | (0.119) | (0.258) | (0.044) | (0.206) |
| Gender Segregation * % Employed in Sector | | 0.440*** | | -1.560*** | | -1.640*** | | 0.394 |
| | | (0.089) | | (0.325) | | (0.404) | | (0.456) |
| Occupational Racial Segregation | -0.060 | -0.189 | -0.223 | -0.116 | -0.042 | 0.014 | -0.161 | -0.144 |
| | (0.164) | (0.158) | (0.147) | (0.155) | (0.284) | (0.284) | (0.298) | (0.291) |
| % College Educated | 14.436*** | 12.211*** | 7.230*** | 6.438*** | 8.125** | 8.263** | 6.556* | 6.692* |
| | (1.068) | (1.319) | (1.778) | (1.777) | (2.584) | (2.535) | (3.080) | (3.158) |
| % Women | -11.903*** | -13.109*** | -4.973*** | -3.578** | -12.862*** | -12.246*** | -3.886* | -4.055* |
| | (1.146) | (1.171) | (1.270) | (1.310) | (2.347) | (2.327) | (1.535) | (1.589) |
| % White | 3.400** | 2.540* | -0.231 | 0.716 | 5.423* | 5.464* | 2.286 | 2.355 |
| | (1.053) | (1.060) | (1.604) | (1.518) | (2.567) | (2.467) | (1.854) | (1.846) |
| Labor Market Characteristics | | | | | | | | |
| Population (logged) | -0.209 | -0.145 | 0.011 | -0.297 | -0.878 | -0.669 | 1.031* | 1.215* |
| | (0.518) | (0.513) | (0.558) | (0.552) | (0.857) | (0.807) | (0.410) | (0.549) |
| % Foreign Born | 9.734 | 11.161* | 14.939* | 14.933** | 26.389** | 19.657* | -3.925 | -4.220 |
| | (5.462) | (5.434) | (5.902) | (5.435) | (9.218) | (8.470) | (3.835) | (4.041) |
| % Aged 25–40 | 8.731 | 9.197 | 2.782 | 1.447 | 8.560 | 10.715 | 19.322*** | 19.589*** |
| | (6.746) | (6.715) | (6.728) | (6.608) | (9.119) | (9.155) | (5.765) | (5.696) |
| % White | -1.329 | -0.405 | -1.232 | -0.648 | 1.357 | 2.158 | 1.341 | 1.314 |
| | (2.135) | (2.144) | (2.935) | (2.535) | (4.252) | (3.892) | (1.752) | (1.766) |
| Unemployment Rate | -5.178 | -5.247* | -12.152*** | -13.002*** | 3.756 | 6.675 | -4.449 | -4.269 |
| | (2.637) | (2.560) | (3.197) | (3.043) | (5.102) | (5.006) | (2.401) | (2.405) |
| % College Educated | 12.870*** | 16.216*** | 32.267*** | 28.083*** | 30.351*** | 24.848*** | 11.810*** | 10.960** |
| | (3.260) | (3.306) | (4.519) | (4.477) | (6.594) | (6.986) | (3.174) | (3.425) |
| % Employed in Export-Intensive Industries | 3.455 | 3.185 | -6.829** | -7.588*** | 3.694 | 1.418 | 1.621 | 1.914 |
| | (2.038) | (2.063) | (2.216) | (2.045) | (4.374) | (4.125) | (1.662) | (1.631) |
| Women's Labor Force Participation | -8.261*** | -7.705*** | -7.089** | -7.774*** | -15.267*** | -13.262*** | -1.597 | -1.041 |
| | (1.791) | (1.774) | (2.119) | (1.981) | (3.978) | (3.894) | (1.246) | (1.464) |
| Service to Goods Employment Ratio | -0.025 | -0.044 | -0.031 | -0.066 | -0.018 | -0.217 | -0.151 | -0.112 |
| | (0.151) | (0.149) | (0.147) | (0.142) | (0.308) | (0.294) | (0.108) | (0.103) |
| Constant | 16.776** | 16.800** | 18.897** | 18.928** | 28.694* | 20.850 | -8.382 | -8.369 |
| | (6.081) | (6.008) | (6.986) | (6.783) | (11.375) | (10.954) | (4.578) | (4.728) |
| Fixed Effects | Year Labor Market Industry | Year Labor Market Industry | Year Labor Market | Year Labor Market | Year Labor Market | Year Labor Market | Year Labor Market | Year Labor Market |
| N | 2148 | 2148 | 716 | 716 | 716 | 716 | 716 | 716 |
| R-sq | 0.924 | 0.925 | 0.811 | 0.821 | 0.533 | 0.549 | 0.761 | 0.762 |

*NOTE*: Percent employed in sector multiplied by 100. Standard errors clustered by labor market, reported in parentheses.

*p<0.05

**p<0.01

***p<0.001

specifically test whether the effects of industry growth and occupational segregation vary between FIRE, tech, and retail trade. To provide a more valid test of the relationship between wage growth and occupational gender segregation, and to test Hypotheses 2 and 3, models 3 through 8 explore the relationship of occupational gender segregation and sector growth to wages using independent equations for each industry.

The second set of models focuses on wages in the FIRE sector. Model 3 reports a significant penalty of occupational gender segregation on wage growth. Labor markets with higher levels of gender segregation in FIRE have experienced less wage growth between 1980 and 2010 ($p <$ .01). This provides support for Hypothesis 2, that occupational gender segregation hinders productivity through preventing the exchange of information and knowledge between workers. Conversely, occupational gender integration provides a boost to wages in FIRE. Areas where segregation in FIRE decreased from high ($d = .6$) to moderate ($d = .4$) are predicted to have growth of $.49 an hour in wages–a raise of about 3% in the average labor market. Among covariates, sector growth is associated with higher wages ($p < .05$), indicating a wage premium for workers in labor markets with dense FIRE industries. The share of workers with a college degree within the industry ($p < .001$) and labor market ($p < .001$) are both associated with higher wages, as are the share of labor market residents that are foreign born ($p < .05$). Women's representation in the industry ($p < .001$) and women's labor force participation rate in the labor market ($p < .01$) are both associated with lower wages, and this is consistent with research on the devaluation of feminized occupations across industries [66,67]. Unemployment is also associated with lower wages ($p < .001$), as well as the share of workers in export intensive industries ($p < .01$). The effects of control variables varied in subsequent models exploring wages in tech and retail trade. The concentration of export-intensive industries and the unemployment rate is associated with lower wages in FIRE, while having a non-significant relationship to wages in both tech and retail trade. Wages in retail trade are positively related to population ($p < .05$) and the concentration of younger working-age residents ($p < .001$), while being unrelated to the share of residents who are foreign born and rates of women's labor force participation (which predicted FIRE and tech wages). Finally, industry racial composition (% white) predicted higher wages in tech ($p < .05$), but not FIRE or retail trade. These varying effects of control variables across industries highlight the importance of independent models predicting wages separately for FIRE, tech, and retail trade, where such complicated dynamics may be efficiently controlled for.

Model 4 tests whether the relationship of FIRE sector growth to wages is moderated by occupational gender segregation. The interaction term between gender segregation and FIRE sector size is significant ($p < .001$), indicating that the separation of women and men into different occupations influences whether FIRE growth will have positive effects on wages. This relationship is illustrated in Fig 3A which plots the predicted coefficient for FIRE sector growth by levels of occupational gender segregation. When occupational gender segregation is low ($d = .35$), FIRE expansion is highly predictive of wage growth ($\beta = .354$, $p < .001$). At moderate levels of gender segregation ($d = .5$), however, the effect of FIRE sector growth reduces by over 60% and becomes non-significant ($\beta = .120$), while at high levels of occupational segregation ($d = .65$), FIRE expansion has a non-significant negative association with wage growth ($\beta = -.113$). These patterns indicate that occupational gender segregation plays a large role in the quality of industry growth. FIRE industry expansion only results in higher wages when women and men work alongside one another in similar occupations. For example, the average labor market with wages of $15 an hour in FIRE is predicted to experience wage growth of 2.4% ($0.35) when FIRE expands by 1 percentage point and occupational segregation is low ($d = .35$), compared to wage growth of .8% ($0.12) when occupational segregation is moderate ($d = .5$). In the context of high occupational gender segregation, FIRE expansion does not increase

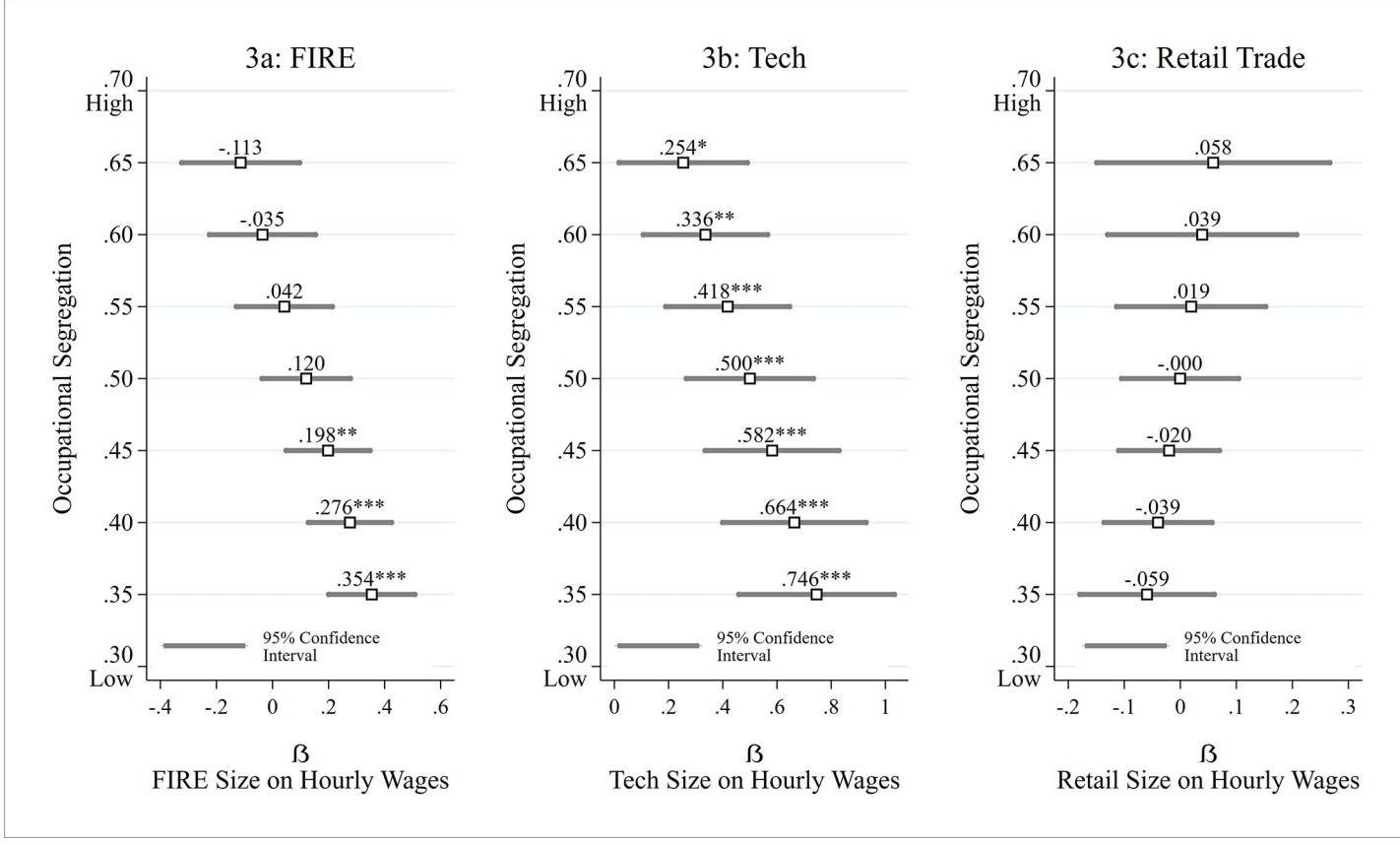

**Fig 3. Coefficient for industry expansion predicting wage growth by levels of occupational gender segregation.**

wages, as productivity is hindered by limited interaction and exchange between women and men workers.

The fifth and sixth models in Table 3 examine the relationship between occupational gender segregation, industry growth, and wages in the tech services sector. Unlike what was observed with regards to the FIRE sector, model 5 reports that occupational gender segregation does not have significant direct effects on wages in the tech services sector. These differences between FIRE and tech may stem from the fact that the tech sector has grown at over twice the rate as FIRE between 1980 and 2010 (see Table 1). Consistent with this, the direct effect of sector growth on wages is over twice as large for tech as FIRE. In the context of such large tech sector growth since 1980, occupational gender segregation may play a less direct role in wages.

While occupational segregation may not have a direct effect on wages in tech, it is possible that it shapes the degree to which sector expansion improves wages (Hypothesis 3). Testing this relationship, model 6 reveals that occupational segregation is a significant moderator of industry size on wages ($p < .001$). In a similar pattern as what was observed with the FIRE sector, tech industry expansion is most strongly associated with wage growth in labor markets with lower levels of occupational gender segregation. In fact, the coefficient for the moderation effect is similar in magnitude as that observed for FIRE, as indicated by a parallel slope in Fig 3B representing the coefficients for tech sector size on wages across levels of occupational gender segregation. When gender segregation is low (d = .35), tech sector growth has the largest positive effect on wages ($\beta = .746$, $p < .001$). At moderate levels of occupational gender segregation (d = .5), the effect of tech sector expansion on wages reduces by 33% ($\beta = .500$,

p < .001), while at high levels of occupational segregation (d = .65), the coefficient for tech expansion reduces by 66% (β = .254, p < .05). It is notable that the effect of tech expansion is positive and significant until occupational gender segregation reaches the high level of .66, which is in the 60th percentile of labor market-years. This pattern provides further insight on why the direct effects of occupational gender segregation observed in model 5 were non-significant. The relationship of occupational segregation to wages in tech primarily occurs in conditions of industry expansion. In other words, tech expansion boosts wages, but the benefits of industry expansion to wages are greater when occupational gender segregation is low, and non-existent when segregation is at very high levels. These results provide further evidence for the benefits of occupational gender integration and the barriers posed by gender segregation to economic growth and productivity. Labor markets with lower levels of gender segregation have more productive tech sectors where information is exchanged and innovation thrives, driving higher wages over time. In the average labor market with tech wages of about $21 an hour, tech expansion of 1 percentage point is predicted to increase wages by 3.6% ($0.75) when segregation is low (d = .35), 2.4% ($0.50) when segregation is moderate (d = .5), and only 1.2% ($0.25) when segregation is high (d = .65).

The last set of models in Table 3 tests the relationship of occupational gender segregation, industry expansion, and wages in retail trade. As a low-skill sector where productivity is less centered on information and innovation, retail trade serves as a counterfactual to FIRE and tech since the effects of occupational gender segregation on information exchange would not provide benefits to productivity in retail trade as it would in FIRE or tech. Focusing on direct effects, model 7 reports that occupational gender segregation is unrelated to wages in the retail trade sector. Testing whether occupational segregation moderates the effects of industry expansion on wages, the interaction term in model 8 is also non-significant (see Fig 3C), showing that industry expansion has similar null effects on wages regardless of levels of occupational gender segregation. These findings indicate that the benefits of women and men working in similar occupations occur primarily through the exchange of information, a driver of growth in high-skill service sectors like FIRE and tech, but not in low-skill sectors such as retail trade.

## Summary of results

Table 4 summarizes the results according to the study's hypotheses. The first hypothesis, that lower levels of occupational gender segregation will predict expansion in FIRE and tech sectors, was supported. The second hypothesis, that gender segregation will have direct effects on wages, was partially supported. Occupational gender segregation had a negative direct effect on wages in FIRE, but not in tech services. This suggests that segregation hinders productivity in FIRE regardless of whether the sector experiences growth or not. Tech productivity,

**Table 4. Summary of results.**

| Hypotheses | Result |
|---|---|
| H1: Lower occupational gender segregation will be associated with expanding finance and tech sectors at the local labor market level. | Supported. The segregation of women and men into different occupations is negatively associated with FIRE and tech sector growth. |
| H2: Lower occupational gender segregation will be associated with wage growth in the high-skill service sectors of finance and tech at the local labor market level. | Partially Supported. Lower levels of occupational gender segregation are directly associated with wage growth in FIRE, but not tech. |
| H3: Finance and tech sector expansion will be associated with wage growth when accompanied by lower levels of occupational gender segregation at the local labor market level. | Supported. FIRE and tech expansion is most strongly associated with wage growth in conditions of low occupational gender segregation. |

meanwhile, is only affected by gender segregation under conditions of changing industry size, indicating that industry expansion is a major contributor to wage growth in tech, but the extent of this wage growth depends on levels of occupational gender segregation. As reported alongside Hypothesis 3 in Table 4, tech expansion was most strongly associated with higher wages in labor markets with lower levels of occupational gender segregation. A similar pattern was observed in FIRE. These trends support Hypothesis 3, suggesting that the integration of women and men in similar occupations is a key ingredient shaping whether industry expansion will be successful in generating an economically vibrant labor market.

## Discussion: A gendered theory of endogenous growth

Results from this study illustrate the dynamic interplay between gender segregation and economic change. The integration of women and men in similar occupations facilitates the exchange of information and creative innovation necessary for the emergence of lucrative financial and tech industries. Absent these conditions, industry expansion results in fewer economic benefits, suggesting that exchange between women and men is necessary for quality industry growth. Previous research at the level of organizations has found that gender diversity improves profits, sales, and productivity [14–19]. The findings from this study suggest that similar patterns occur at the broader level of local labor markets.

Collectively, these results indicate a need for a theory of labor market economic development that explicitly considers the value of gender equity. Research on local culture and social capital has emphasized how the exchange of diverse information and ideas can stimulate innovation and growth [23,25,36,41]. By recognizing gender as a social structure [43] that shapes the experiences, perspectives, and knowledge of women and men in systematically different ways [44], a *gendered theory of endogenous economic growth* provides a framework for viewing occupational gender integration as a vital asset for economic development through its ability to increase the flow of knowledge and information within labor markets. Consistent with previous theories of endogenous growth, this framework proposes that the potential for innovation in a labor market is equal to the product of the knowledge contained by local workers [31,34,36]. Where this new framework departs from previous theories, however, is in the explicit attention to gender integration as an asset, and gender segregation as an impediment, to the utilization of worker knowledge.

By virtue of possessing different structural locations in society [43,44], women and men often have different perspectives and standpoints. In a highly segregated labor market where women and men work predominantly in separate occupations, there is little opportunity for them to learn from one another or share their differing views. Because creativity and innovation is facilitated by the exchange of diverse perspectives, gender segregation limits workers' productivity by homogenizing not only the types of workers in particular jobs, but also limiting the ideas, knowledge, and perspectives that are exchanged among workers and workforce leaders. A more gender-integrated labor force where women and men work alongside each other in the same occupations, meanwhile, takes advantage of the fact that women and men have diverse experiences and ideas by virtue of their different structural locations in society. Information heterogeneity is maximized in such an environment, and along with it the ability for workers to solve new problems and generate new innovations [33,36]. Furthermore, the exchange of information between diverse actors, such as women and men, may have a multiplicative effect where the creative capacity of both is improved from their mutual exchange. In other words, workers are individually enriched through the information they gain from diverse colleagues, and through this increased capacity they contribute to economic growth in their labor market.

## Limitations and future directions

This study has established a relationship between occupational gender segregation and economic growth. There are, however, limitations to the analytic approach taken here. First, the estimation strategy is unable to control for unobserved changing characteristics of industries and labor markets, as well as account for potential endogeneity where characteristics of tech and FIRE sectors affect levels of gender segregation. To account for these factors, I used retail trade as a counterfactual case that would be affected by similar confounding variables as FIRE and tech, but should not be influenced in the same way by occupational gender segregation since retail trade relies less heavily on innovation and information. This approach added further support to this study's findings. Future research may build from the present study using alternative strategies designed to address issues related to omitted variables and endogeneity. While I was unable to identify a suitable instrumental variable, future research using different sources of data may take this approach in testing the findings presented here.

I have relied upon previous research and theory in interpreting observed relationships as evidence that the exchange between women and men occurring in gender-integrated labor markets facilitates the sharing of information and knowledge leading to overall improvements in economic growth. Yet, alternative mechanisms may also be at play, which I am unable to test in the data used here. It is possible, for example, that skilled women workers are moving to labor markets with lower levels of gender segregation. Thus, rather than facilitating the exchange of information to produce growth endogenously, occupational gender integration may be attracting highly-skilled workers and boosting growth through exogenous mechanisms. Future research using alternative survey data or qualitative methods may be better positioned to investigate the mechanisms by which gender segregation hinders, and integration boosts, local economies.

## Conclusion

The segregation of women and men into different occupations poses an economic barrier to local labor markets. Industry expansion in finance and tech sectors only improves local economies when there are lower levels of gender segregation. For labor markets to be competitive in the new economy, they need to reduce levels of segregation. Improving the gender integration of occupations will help unlock the creative and innovative potential of labor markets through facilitating the exchange of diverse ideas and perspectives between women and men.

## Supporting information

**S1 Appendix. Matching PUMS respondents to labor markets.**
(DOCX)

**S2 Appendix. Fixed effects models predicting wages at 25th, 50th, and 75th percentiles.**
(DOCX)

## Acknowledgments

The author would like to thank Barbara Risman, William Bielby, Erin Cech, Leslie McCall, Moshe Semyonov, and the three anonymous reviewers for feedback on earlier versions of this article.

## Author Contributions

**Conceptualization:** William J. Scarborough.

**Data curation:** William J. Scarborough.

**Formal analysis:** William J. Scarborough.

**Investigation:** William J. Scarborough.

**Methodology:** William J. Scarborough.

**Project administration:** William J. Scarborough.

**Resources:** William J. Scarborough.

**Software:** William J. Scarborough.

**Supervision:** William J. Scarborough.

**Validation:** William J. Scarborough.

**Visualization:** William J. Scarborough.

**Writing – original draft:** William J. Scarborough.

**Writing – review & editing:** William J. Scarborough.

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
