## [Decision Letter · Decision Letter 0]

17 Oct 2019

PONE-D-19-24965

Gender segregation of managerial/professional occupations and economic growth in U.S. labor markets, 1980 through 2010

PLOS ONE

Dear Dr. Scarborough,

Thank you for submitting your manuscript to PLOS ONE. After careful consideration, we feel that it has merit but does not fully meet PLOS ONE’s publication criteria as it currently stands. Therefore, we invite you to submit a revised version of the manuscript that addresses the points raised during the review process.

All the referees are labor / urban economists and are very familar with IPUMS data. Their recommendations are split from minor revision, major revision, to rejection. Since your study may generate some media attention once it is published, it is important to establish a convincing conclusion. Please address the referees' concerns as much as you can. I also want to add a few more comments below to help push your study a bit further.

The phrase “U.S. labor markets” in the title is a bit confusing. It could refer to the whole labor market in the U.S. or to the many local labor markets that you are talking about. Perhaps using “U.S. local labor markets” is clearer?In the introduction, please add a few sentences to state your estimation strategy, that is, how your fixed effect model helps identify the causal effect of gender occupation segregation on growth.You focus on only managerial and professional occupations and finance and tech industries. This poses a question to readers whether gender segregation is occupation specific, and whether its effect is industry specific. You did not provide enough justification. Given you have all the data, you can check if gender segregation prevails in all or most of occupations. You can also estimate your model including more industries and use industry fixed effects to control for industry heterogeneity to see if the effect of gender segregation is general. At least you should provide more reasons to justify your current focus. Also for example for Table 2, you can pool all data together and estimate one model with industry dummies, this will show a general pattern of your results. You can add this as the 3^rd^ column.Female labor force participation is an important omitted variable. Suppose female labor force participation rate is low in a labor market, the dissimilarity index will have a higher value simply because fewer females are in the labor market. Consider the importance of racial segregation, you may also need to control for black-white dissimilarity index. I notice you used share of whites as a control variable, perhaps black-white dissimilarity index is a better control.In the analytic approach section, it’s better to explicitly write down the econometric models you estimate, followed by the definitions of dependent and independent variable, and then the estimation strategy. Besides the omitted variable issue, you face the spatial sorting issue: if skilled female workers are more likely to move to labor markets with tolerance and gender equality, you will see a correlation between less gender segregation and higher industry growth. Although you did not (or cannot) deal with this issue, you should at least acknowledge this limitation. (For your reference, in the literature of racial segregation, some studies use instrumental variables for racial segregation, see for example*, *Ananat, Elizabeth Oltmans. *2011.*
*"The Wrong Side(s) of the Tracks: The Causal Effects of Racial Segregation on Urban Poverty and Inequality."*
*American Economic Journal: Applied Economics*, *3 (2): 34-66**.)*There are some studies on urban cultural diversity and innovation or productivity that you may cite. For example: Gianmarco I.P. Ottaviano, Giovanni Peri, 2005, Cities and cultures, Journal of Urban Economics 58 (2), 304-337. 

We would appreciate receiving your revised manuscript by Nov 25 2019 11:59PM. To enhance the reproducibility of your results, we recommend that if applicable you deposit your laboratory protocols in protocols.io, where a protocol can be assigned its own identifier (DOI) such that it can be cited independently in the future. For instructions see: http://journals.plos.org/plosone/s/submission-guidelines#loc-laboratory-protocols

We look forward to receiving your revised manuscript.

Kind regards,

Shihe Fu, Ph.D.

Academic Editor

PLOS ONE

**Journal Requirements:**

2. We note that  Figure(s) 1 in your submission contain [map/satellite] images which may be copyrighted. All PLOS content is published under the Creative Commons Attribution License (CC BY 4.0), which means that the manuscript, images, and Supporting Information files will be freely available online, and any third party is permitted to access, download, copy, distribute, and use these materials in any way, even commercially, with proper attribution. For these reasons, we cannot publish previously copyrighted maps or satellite images created using proprietary data, such as Google software (Google Maps, Street View, and Earth). For more information, see our copyright guidelines: http://journals.plos.org/plosone/s/licenses-and-copyright.

a) You may seek permission from the original copyright holder of Figure(s) [#] to publish the content specifically under the CC BY 4.0 license.  

**Comments to the Author**

1. Is the manuscript technically sound, and do the data support the conclusions?

Reviewer #1: Partly

Reviewer #2: Yes

Reviewer #3: Yes

2. Has the statistical analysis been performed appropriately and rigorously? 

Reviewer #1: No

Reviewer #2: Yes

Reviewer #3: Yes

3. Have the authors made all data underlying the findings in their manuscript fully available?

Reviewer #1: Yes

Reviewer #2: Yes

Reviewer #3: Yes

4. Is the manuscript presented in an intelligible fashion and written in standard English?

Reviewer #1: Yes

Reviewer #2: Yes

Reviewer #3: Yes

5. Review Comments to the Author

Reviewer #1: This paper studies the impact of gender segregation on economic growth in the U.S. In contrast to previous studies that mostly using organization-level data (e.g., firm-level), this paper conducts empirical analysis at the local labor market level, which is defined as labor sheds. It attempt to measure managerial/professional occupational gender segregation at the local labor market level and provides two major empirical results. First, a higher level of segregation is associated with reduction in the employment in finance and technological service sectors. Second, a higher level of segregation is associated with worse economic well-being measured as labor market median wages, household income, and business growth.

....

The rest of the report please see the attachment.

Reviewer #2: This paper examines the relationship between gender segregation of managerial/professional occupations and economic well-being at the labor market level using micro census data harmonized in the Integrated Public Use Microdata Series (IPUMS) from 1980 to 2010. In the study, labor markets are defined as BEA labor sheds, and economic well-being are measured by (1) median hourly wage, (2) median household income, and (3) number of business establishments. The focal variable, gender segregation of management/professional occupations, is measured by the index of dissimilarity. The study finds a negative association between gender segregation and the expansion of finance and technology sectors, as well as a negative association between gender segregation and overall labor market economic well-being.

Major comments:

1) The author highlighted the unit of analysis or the level of aggregation (i.e. labor market level) as one of the contributions of this current study, as a strong relationship between gender diversity and firm performance has been established in the sociology literature (i.e. firm level or organizational level). In my opinion, the appropriate level of aggregation for each study depends on the underlying mechanism at play and its corresponding measurement. Given that the major channel the author hypothesized is the exchange of new ideas and unique information, it is not clear why labor market would be a more appropriate unit of analysis compared with firm or industry in the same location. In the discussion section of the paper, the author also recommended future studies to use more detailed level units of analysis to investigate interactions between workers.

Related to the level of aggregation, I have the following comments about the focal variable:

a) There seems to be a mismatch in the gender segregation measure and the hypothesized channel. From reading the paper, I understand that the author focused on finance and tech industries due to importance of the exchange of ideas and information, which would be hindered by gender segregation. However, the gender segregation measure is constructed by all industries in a given labor market-year, rather than the finance and tech industries alone.

b) It is not clear how the gender segregation index is constructed. The author stated that it is the index of dissimilarity. However, given the number of managerial/professional occupations included in the study (see page 37), more detailed steps of the index calculation should be presented clearly.

2) Even though the author was not arguing for causality in this paper, I still think the potential omitted variable bias should be discussed more carefully. Labor market fixed effects can account for permanent features of a region. However, there can still be some time-varying characteristics of a region that is positively associated with a more diverse labor force and a booming local economy with strong finance and tech sectors. For example, the skill intensity of the labor force (i.e. share of college educated workers) could be positively correlated with both a booming local economy with strong finance and tech sectors and a more diverse labor force, and it is changing over time. Moreover, skill intensity is also important to the hypothesized channel of idea exchange and information sharing. In fact, in Table 3 on page 19, the estimated coefficients on population are positive and statistically significant across all specification. In the urban economics literature, a positive correlation between city size and skill intensity has been well established.

Minor point:

3) It is not clear how the standard errors are calculated. In my opinion, standard errors in this type of study should be clustered at the appropriate level of aggregation, for example, the labor market level.

Reviewer #3: This paper uses 1980, 1990, and 2000 census data with the 2010 American Community Survey 5-year microdata sample to form a panel data at the labor-shed level to empirically study the relationship between labor market levels of gender segregation among managerial/professional occupations and economic growth as well as the effect on labor market economic well-being (median hourly wages, median household income, and growth in the number of business establishments). The results show that a lower level of gender segregation helps expand in the finance and the tech sectors. And, in general, a lower level of gender segregation increases median hourly wages, median household income, and business growth.

This paper is interesting and has important policy implications. The paper is well-written. I recommend accepting this paper under the condition that the following issues are addressed.

1. Does the sample used in the regression analysis include immigrants? In the US, immigrant ratios in local labor markets could be possibly correlated with gender segregation and determine economic growth and economic well-being, resulting in biased and inconsistent results. The two-way fixed effects model may not address this issue.

2. Table 1 reports the change in labor market characteristics, not the level. I suggest addionally report the level of the key variables can offer better information about the data used.

3. Why use median hourly wages and median household income, not the mean? The paper does not give a reason. Is it because large outliers in the two variables so the paper uses medians? If using medians for the dependent variables, why not doing quantile regressions and show other interesting results such as quartiles or deciles?

4. The paper did not discuss the economic significance of the estimated coefficients. That is, for example on page 17, do the sizes of the estimates make sense? At least some discussion will be helpful.

5. What kind of standard error does the paper use? I did not find it. The tables simply report “Standard errors in parentheses”. Since the paper uses panel data regressions, reporting clustered standard errors helps reduce the efficiency concern.

6. Tables 2 and 3 should report R2.

7. Models in Table 3 use interaction terms and report all estimated coefficients, which is hard to interpret the results regarding the effect of gender segregation on each of the outcome variables. I suggest to additionally report marginal effects in the table. A better way is to draw a plot to show how the effect changes at different levels of gender segregation.

6. PLOS authors have the option to publish the peer review history of their article (what does this mean?). If published, this will include your full peer review and any attached files.

Reviewer #1: No

Reviewer #2: No

Reviewer #3: No

---

## [Author Response · Author response to Decision Letter 0]

11 Nov 2019

Response to Editor’s and Reviewers’ Comments:

I would like to thank the three reviewers and the editor for their encouragement about the paper and also for their constructive comments. I have revised the paper to address each comment and describe these revisions in detail below. I believe the paper is much improved after incorporating feedback, and I would be happy to make additional revisions that you believe would further improve the paper.

Similar Comments from Editor and Reviewers:

1. Both the editor and Reviewer 1 suggested that I specify my focus on “local” labor markets, particularly in the title of the manuscript.

Response: I have changed the title of the manuscript to reflect my focus on local labor markets. I have also made changes throughout the manuscript to specify that I’m focusing on local geographic labor markets.

2. The editor, Reviewer 1, and Reviewer 2 each commented on the restricted use of managerial/professional occupational segregation in the original manuscript. Both the editor and Reviewer 1 suggested that I use occupational gender segregation across all occupations. Relatedly, both Reviewer 1 and Reviewer 2 suggested that I add more detail to the measurement of occupational gender segregation.

I have changed my focal predictor to reflect occupational gender segregation across all occupations (as opposed to just managers/professionals) for each industry-labor market-year. These changes are reflected throughout the manuscript, and are described in detail on pages 12 and 13 where I discuss the measurement of occupational gender segregation.

Following Reviewer 1 and 2’s suggestion, I have also added more detail to the measurement of occupational gender segregation. As discussed in the text (pages 12 and 13), I use one of the most common measures of segregation, the index of dissimilarity. Equation 2 (page 12) provides detail on how the index is calculated, and I discuss its interpretation also on page 13.

3. The editor and all three reviewers offered suggestions on additional control variables, including women’s labor force participation, percent with a college degree, percent foreign born, racial segregation, and degree of international trade.

Thank you for the suggestions for additional controls. I have added all recommended covariates. These are discussed in the analytic strategy section on page 13 and reflected in the results in Tables 2 and 3.

4. Reviewers 1 and 3 suggested that I report labor market characteristics for each year in the descriptive section of the results instead of focusing on change.

I have replaced Table 1 with a new table reporting descriptive statistics for each year included in the study.

5. Reviewers 1 and 3 suggested that I discuss the magnitude of coefficients in the results and discuss their economic significance.

Throughout the results section, I have added more interpretation to regression coefficients in order to highlight their economic significance. For example, I now discuss how a change in gender segregation from highly segregated to moderately segregated predicts change in hourly wages. These revisions are present on pages 19 and 23-25.

6. Reviewers 2 and 3 suggested that I specify how standard errors are calculated, with both suggesting that standard errors be clustered by labor markets.

Standard errors are clustered by labor markets. I now report this on page 14 and in Tables 2 and 3.

7. The editor and Reviewer 1 suggested additional citations to studies on urban cultural diversity and productivity.

Thank you for pointing me to the additional citations. They fit well with the framing of the paper around the merits of local diversity. I have added them in the front end of the paper, most noticeably on pages 5 and 6.

Additional Comments from the Editor

1. The editor suggested I add a few sentences in the introduction to state my estimation strategy.

I have added text to page 4 stating my use of fixed effects regression models to estimate the relationship of occupational gender segregation to economic outcomes. Following the suggestion of Reviewer 1, (discussed below), I also state my use of retail trade as a counterfactual industry where the hypothesized effects of occupational gender segregation should not be observed as they are in the high-skilled service industries of finance and tech.

2. The editor suggested that I include a pooled model in Table 2 that includes all industries with fixed effects for industry to identify the general effects of occupational gender segregation.

I have added a pooled model to Table 2, including fixed effects for industry. As the editor suggested, this provides insight on the general effects of gender segregation across industries. These changes are reflected in Table 2, on page 19, as well as in my discussion of the estimation strategy on pages 10-12.

3. The editor suggested I explicitly write the econometric models estimated, followed by variable definition.

I have rewritten the analytic strategy section in accordance with the editor’s recommendation. The econometric fixed effects equations are now written explicitly, with corresponding variable definitions following. These changes are reflected on pages 11 through 14.

4. The editor noted an additional limitation to the study: That it cannot identify whether the effects of gender segregation are due to growth through the sharing of information among workers that boosts productivity, or through growth occurring when highly skilled women workers move to labor markets with lower levels of occupational segregation. The editor advised me to acknowledge this limitation, since it is unlikely to be resolved given the current data.

As the editor predicted, I am unable to identify the role of spatial sorting in my results, given the present data. Therefore, I now state this as a limitation on page 30.

5. The editor provided references to studies that have used instrumental variables to study racial segregation as suggestions of how a similar approach may be used to study occupational segregation.

Thank you for this useful reference. Ananat’s use of railroad tracks was a very creative application of instrumental variables to study racial segregation. In my revised manuscript, I have noted my attempts to identify a suitable instrument (pages 11 and 29). Ultimately, I was unable to identify an instrument. However, the suggestion from Reviewer 1, to use a low-skill service industry as a counterfactual (discussed below), helped bolster the claims I make in the paper by showing how the benefits of occupational gender integration to economic growth are experienced primarily in high-skill service industries (finance and tech) where information exchange is more important, while being non-existent in low-skilled service industries (retail trade) where information exchange plays a smaller role.

Additional Comments from Reviewer 1

1. Reviewer 1 suggested that I add a map indicating levels of occupational gender segregation across local labor markets to add further descriptive detail to this variable.

I have added Figure 2 to the text, illustrating levels of occupational gender segregation in finance, tech, and retail trade industries across U.S. local labor markets in 2010. I discuss the geographic variation of occupational gender segregation on pages 16 and 17.

2. Reviewer 1 expressed surprise that FIRE increased only by .6 percentage points between 1980 and 2010, when a BLS report showed an increase of 7% between 1979 and 2001.

Thank you for providing the link to the BLS report. The 7% figure referenced from the report refers to FIRE growth as a proportion of its 1979 size (i.e. ∆FIRE from 1979 to 2001/1979 FIRE size), whereas change indicated in Table 1 refers to the percentage point difference in the size of the FIRE sector (i.e. 2010 FIRE – 1980 FIRE). When translating the information in Table 1 to percentage growth (∆FIRE from 1980 to 2010/1980 FIRE size), there is an 11% increase in the size of FIRE industries across labor markets, which is consistent with the 7% increase Reviewer 1 cited as occurring between 1979 and 2001. Additionally, further reading in the BLS report referenced by Reviewer 1 (https://www.bls.gov/opub/mlr/2016/article/current-employment-statistics-survey-100-years-of-employment-hours-and-earnings.htm) indicates that the size of the FIRE sector was 4.6 percent of employment in 1949 and 5.7 percent of employment in 2015 – consistent with the figures reported in Table 1.

3. Reviewer 1 pointed out the potential for reverse causation between occupational gender segregation and economic outcomes. The reviewer suggested that, if unable to identify a valid instrument, I acknowledge this limitation. The reviewer also suggested that I examine a low-skill industry where the sharing of information would be less related to economic growth as a counterfactual to the high-skill industries of finance and tech.

Thank you for these helpful suggestions to my methods. As suggested, I have noted the potential for endogeneity related to reverse causality (pages 11 and 29). I have also followed Reviewer 1’s suggestion to use a low-skill service industry as a counterfactual case. I use retail trade as a counterfactual. I believe this suggestion greatly improved the validity of my findings because the counterfactual industry would be similarly affected by omitted variables as finance and tech, but would not be similarly affected by the mechanism of interest – how occupational gender segregation facilitates information exchange to improve economic output.

I discuss the use of a counterfactual on pages 4, 11, and throughout the results section. As expected if occupational segregation operates through the hypothesized mechanism, no significant effects were observed in retail trade, while they were present in finance and tech.

4. Reviewer 1 suggested that I focus on wages as a primary indicator of economic productivity.

Following Reviewer 1’s suggestion, I now focus primarily on hourly wages as an indicator of economic productivity in finance, tech, and retail trade sectors.

5. Reviewer 1 recommended that I discuss the difference between labor sheds and metros / commuting zones.

I have added text to page 9 to specify how labor sheds offer an advantage over metro areas in delineating commuting zones where individuals live and work.

6. Reviewer 1 suggested I split Table 3 into two tables corresponding to hypotheses 2 and 3.

In this revision, I did not generate an additional table because I reduced my focus from three economic outcomes (wages, household income, establishment growth) to wages as an indicator of productivity (following Reviewer 1’s suggestion from above). By pairing direct and moderating effects in Table 3, I am able to succinctly interpret the full association of occupational gender segregation to wages for each industry. If the reviewer continues to feel that splitting Table 3 into two tables would be ideal, I would be very willing to do so.

Additional Comments from Reviewer 2

1. Reviewer 2 recommended further justification for the use of labor sheds as local labor markets.

On page 9, I have added further justification for the use of labor sheds. In particular, I state that labor sheds are the best measure for commuting zones. Because the hypothesized effects of occupational gender segregation on economic outcomes occurs through exchange between workers, labor sheds are an ideal unit of analysis by virtue of delineating the boundaries where employees work and reside. Further following the reviewer’s suggestion, I also focus on specific industries within labor markets (pages 11-13) to engage more precisely with the mechanisms by which occupational gender segregation affects economic outcomes.

2. Reviewer 2 suggested that I discuss limitations related to omitted variable bias.

As the reviewer suggested, my use of fixed effects models is unable to control for unobserved time variant characteristics. I state this limitation on pages 11 and 29.

 Additional Comments from Reviewer 3

1. Reviewer 3 suggested examining other levels of the wage distribution besides the median to test whether results remain consistent.

I have added Appendix B to the manuscript reporting results of models predicting wages at the 25th, 50th, and 75th percentile. Findings are generally consistent with those reported in the main text focusing on the median. However, effects are strongest at the 75th percentile. This is consistent with the fact that this part of the wage distribution corresponds to high-skill workers most likely to benefit from the exchange of information occurring in low-gender segregated environments.

Because the findings were consistent across the wage distribution, and because the study focuses on the general relationship of occupational gender segregation to economic productivity, rather than the wage distribution within industries, I discuss median wages in the main text and report other aspects of the wage distribution in Appendix B.

2. Reviewer 3 recommended that the r-squared for each model be added to Tables 2 and 3.

I have added the r-squared to each model in Tables 2 and 3. Thank you for your attention to detail.

3. Reviewer 3 suggested that I visualize interactions by illustrating how the effect of industry growth changes depending on levels of occupational gender segregation.

I have added Figure 3 which reports the predicted coefficient for industry growth across varying levels of occupational segregation.

---

## [Decision Letter · Decision Letter 1]

4 Dec 2019

PONE-D-19-24965R1

Occupational gender segregation and economic growth in U.S. local labor markets, 1980 through 2010

PLOS ONE

Dear Dr. Scarborough,

Thank you for submitting your manuscript to PLOS ONE. After careful consideration, we feel that it has merit but does not fully meet PLOS ONE’s publication criteria as it currently stands. Therefore, we invite you to submit a revised version of the manuscript that addresses the points raised during the review process.

All reviewers are happy with your revision. One referee still has a few minor comments. In addition, I think one statement is incorrect. On p.11, the first paragraph, you wrote "While instrumental variables are often used to address such problems of endogeneity, this approach was not possible here, where economic outcomes and occupational gender segregation are interrelated in ways that prevented identifying an appropriate instrument."  It is because  dependent variable and key independent variable are interrelated that we need to search for an IV. You could simply state that in your setting it is difficult to find an approriate instrumental variable. (It is always not easy to find a good IV, as a matter of fact.)

We would appreciate receiving your revised manuscript by Jan 18 2020 11:59PM. To enhance the reproducibility of your results, we recommend that if applicable you deposit your laboratory protocols in protocols.io, where a protocol can be assigned its own identifier (DOI) such that it can be cited independently in the future. For instructions see: http://journals.plos.org/plosone/s/submission-guidelines#loc-laboratory-protocols

We look forward to receiving your revised manuscript.

Kind regards,

Shihe Fu, Ph.D.

Academic Editor

PLOS ONE

Reviewers' comments:

Reviewer's Responses to Questions

**Comments to the Author**

1. If the authors have adequately addressed your comments raised in a previous round of review and you feel that this manuscript is now acceptable for publication, you may indicate that here to bypass the “Comments to the Author” section, enter your conflict of interest statement in the “Confidential to Editor” section, and submit your "Accept" recommendation.

Reviewer #1: All comments have been addressed

Reviewer #2: All comments have been addressed

Reviewer #3: All comments have been addressed

2. Is the manuscript technically sound, and do the data support the conclusions?

Reviewer #1: Yes

Reviewer #2: Yes

Reviewer #3: (No Response)

3. Has the statistical analysis been performed appropriately and rigorously? 

Reviewer #1: Yes

Reviewer #2: Yes

Reviewer #3: (No Response)

4. Have the authors made all data underlying the findings in their manuscript fully available?

Reviewer #1: (No Response)

Reviewer #2: Yes

Reviewer #3: (No Response)

5. Is the manuscript presented in an intelligible fashion and written in standard English?

Reviewer #1: Yes

Reviewer #2: Yes

Reviewer #3: (No Response)

6. Review Comments to the Author

Reviewer #1: (No Response)

Reviewer #2: I believe the authors have addressed most of my comments carefully. In particular, I like the contrasting results between higher-skilled versus lower-skilled industries.

Reviewer #3: (No Response)

7. PLOS authors have the option to publish the peer review history of their article (what does this mean?). If published, this will include your full peer review and any attached files.

Reviewer #1: No

Reviewer #2: No

Reviewer #3: No

---

## [Author Response · Author response to Decision Letter 1]

17 Dec 2019

Response to Editor and Reviewer Comments:

Thank you for the opportunity to revise my manuscript and for the valuable comments. I have followed all suggestions, and believe the article has further improved. These changes are summarized below.

Comments from Editor

1. The editor suggested I correct a statement on page 11 referring to the challenge of identifying a suitable instrumental variable.

Response: Thank you for identifying the problem with this statement. I have followed the suggestion and stated, as suggested, that I was unable to find an appropriate instrumental variable in the case of this study.

Comments from Reviewer:

1. The reviewer noted that the coefficient in the pooled model in Table 2 for the effect of occupational gender segregation was not intuitive, since it was much larger in magnitude than what was observed in subsequent equations independently modelling the effect by industry. The reviewer suggested that I clarify why this pattern is observed.

Response: I have added text to page 21 noting the larger coefficient in the pooled model compared to the models for FIRE and tech and providing an explanation. While the pooled model controls for unobserved characteristics of industries through a fixed effect term, it does not account for the varying effects of controls across industries. The subsequent equations run separately by industry allow for this level of detail. For example, the size of export-intensive industries has no effect on FIRE growth, a positive effect on Tech growth, and a negative relationship with growth in retail trade. Controlling for these types of varying effects of controls mediated the coefficient for occupational segregation from what would be expected given its size in the pooled model. Nonetheless, the coefficients remain significant and substantial. 

2. The reviewer suggested that I provide more discussion of why the direct effects of occupational gender segregation on wage growth were significant for FIRE but not for tech.

I have added text to pages 27 and 28 discussing why there were differing direct effects of occupational gender segregation on wages in FIRE and tech industries. While both FIRE and tech industries grew between 1980 and 2010, tech grew at over twice the rate as FIRE. In the context of such large growth, industry expansion played a more prominent role in wages in tech. This is consistent with the fact that the coefficient for industry expansion on wages is over twice as large for tech as FIRE. Furthermore, the results illustrated in Figure 3b show that the effect of occupational gender segregation on wages in tech occurs primarily in the context of industry expansion (noted on page 28).

3. The reviewer suggested that I include models reporting results for the pooled sample in Table 3, mirroring the approach done in Table 2.

I have added two additional columns to Table 3 reporting results for a pooled model including all industries and with an added industry fixed effect. Discussion of these results has been added to page 24, where I also note the limitations of the pooled model, which is unable to examine varying effects of predictors across industries. This provides a suitable set-up for subsequent models that more directly test the industry-specific hypotheses using independent equations that control for the varying effects of control variables and focus on the effects of occupational gender segregation in FIRE, tech, and retail trade.

4. The reviewer recommended that I clarify the hypotheses to specify measurement at the local labor market level.

Thank you for the specific suggestion of language that would clarify the hypotheses. I have directly incorporated this recommendation to each hypothesis, located on pages 8 and 9.

---

## [Editor Report · Decision Letter 2]

26 Dec 2019

Occupational gender segregation and economic growth in U.S. local labor markets, 1980 through 2010

PONE-D-19-24965R2

Dear Dr. Scarborough,

We are pleased to inform you that your manuscript has been judged scientifically suitable for publication and will be formally accepted for publication once it complies with all outstanding technical requirements.

With kind regards,

Shihe Fu, Ph.D.

Academic Editor

PLOS ONE
---

## [Editor Report · Acceptance letter]

6 Jan 2020

PONE-D-19-24965R2 

Occupational gender segregation and economic growth in U.S. local labor markets, 1980 through 2010 

Dear Dr. Scarborough:

I am pleased to inform you that your manuscript has been deemed suitable for publication in PLOS ONE. Congratulations! Your manuscript is now with our production department. 

With kind regards,

on behalf of

Dr. Shihe Fu 

Academic Editor

PLOS ONE